# Range geography and temperature variability explain cross-continental convergence in range and phenology shifts in a model insect taxon

**Catherine Sirois-Delisle, Susan CC Gordon*, Jeremy Kerr**

University of Ottawa, Ottawa, Canada

## eLife Assessment

The article presents **important** findings on the impact of climate change on odonates, integrating phenological and range shifts to broaden our understanding of biodiversity change. The study leverages extensive natural history data, offering a **convincing** analysis of temporal trends in phenology and range limit and their potential drivers.

**\*For correspondence:**
susan.ccgordon@gmail.com

**Competing interest:** The authors declare that no competing interests exist.

**Abstract** Climate change may introduce conditions beyond species' tolerances; to survive, species must avoid these extremes. Phenological shifts are one strategy, as species move their activity or life-history events in time to avoid extreme conditions. Species may also shift in space, moving their ranges poleward to escape extremes. However, whether species are more likely to exhibit one or both strategies, and whether this can be predicted based on a species' functional traits, is unknown. Using a powerful macroecological dataset of European and North American odonate observations, we assessed range and phenology shifts between two time periods (1980–2002 and 2008–2018) to measure the strength and direction of the association between responses. Species with the greatest poleward range shifts also showed the largest phenological shifts toward earlier annual activity periods, with half of all species shifting in both space and time. This response was consistent across continents, despite highly divergent land use and biogeographical histories in these regions. Surprisingly, species' range and phenology shifts were not related to functional traits; rather, southern species shifted their range limits more strongly, while increasing temperature variability hindered range shifts. By reducing risk through phenological shifts, the resulting larger populations may be more likely to disperse and expand species' ranges. Species shifting in both space and time may be more resilient to extreme conditions, although further work integrating abundance data is needed. We also identified a small number of species (approximately 10%) that failed to shift at all; these species are likely to be particularly vulnerable to climate change and should be prioritized for conservation intervention.

## Introduction

Climate change alters climatic means and increases the frequency of extreme weather events, exposing species to conditions outside of their tolerances and often leading to population declines (*Goulson, 2019*; *IPCC, 2021*). Species may avoid extreme conditions by dispersing to new areas where conditions pose fewer weather-related challenges, often leading to poleward range expansion (*Davis et al., 2005*; *Lawlor et al., 2024*). Species' biological timing could also shift, through adaptation or phenotypic plasticity, with earlier warming advancing the timing of early-season activities

**Figure 1.** Representation of temporal and geographical limits characterizing the ecological niche of a hypothetical odonate species. Points show 250 individuals according to their Julian day of emergence, latitudinal position, and temperatures to which each individual is exposed. Points represent historical observations (T1), plus signs show observations following a shift toward earlier emergence dates after warming (T2a), and triangle symbols show observations following a shift toward higher latitudes after warming (T2b). Species could also shift both range and phenology in response to warming (T2c). Warm and cool colors show hot and cold temperatures, respectively.

and life-history events (*Davis et al., 2005*; *Hällfors et al., 2024*; *Novella-Fernandez et al., 2023*; *Parmesan and Yohe, 2003*). Both species' geographical ranges and seasonal timing depend strongly on climate and habitat conditions, with shifts in space and time permitting species to remain within the limits of their ecological niches (*Figure 1* ; *Chen et al., 2011*; *Engelhardt et al., 2022*; *Grewe et al., 2013*; *Menzel et al., 2006*; *Parmesan and Yohe, 2003*). This allows populations to grow, despite changing environments, and reduces the risk of climate debt and extinction (*Devictor et al., 2012*; *Franks et al., 2018*; *Lustenhouwer et al., 2018*; *Saino et al., 2011*; *Souza et al., 2023*; *Urban, 2015*).

Positive population trends can be stronger in species that shift both their range and phenology (*Hällfors et al., 2024*). Greater phenological plasticity under warmer spring temperatures may increase reproductive success, leading to greater population growth and range expansions (*Macgregor et al., 2019*), as positive or stable trends in species abundance and habitat availability are essential for range shifts (*Mair et al., 2014*; *Platts et al., 2019*). However, there could also be a tradeoff between pheno-logical and geographical shifts (*Amano et al., 2014*; *Hassall, 2015*; *Socolar et al., 2017*). Species with greater dispersal abilities may have less need for phenological shifts as they track their climatic niche through space, while weaker dispersers may be confronted with greater selective pressure to shift phenology within their range (*Amano et al., 2014*; *Hassall, 2015*; *Socolar et al., 2017*). Since range shifts can also result from extirpations at species' trailing range edge (*Parmesan et al., 1999*), greater phenological shifts may mitigate the need for range shifts, as species better tolerate new climatic conditions. Cross-continental studies report converging effects of climate change on species' range shifts and abundances (*Neate-Clegg et al., 2024*; *Stephens et al., 2016*), including among insects (*Kerr et al., 2015*; *Neate-Clegg et al., 2024*; *Pinkert et al., 2022*), but potential relation-ships between phenological and geographic responses have not yet been investigated at continental scales. Functional traits, such as dispersal, may determine species' spatial and temporal responses to climate change (*Chen et al., 2011*; *Kharouba et al., 2009*; *Schuetz et al., 2019*; *Zografou et al., 2021*). However, these relationships are inconsistent across taxa and regions, and cross-continental tests have not been attempted (*Angert et al., 2011*; *Buckley and Kingsolver, 2012*; *Estrada et al.,*

2016; *MacLean and Beissinger, 2017*). Geographic locations and environmental characteristics of species' ranges may also predict range shifts, as animal species with high latitude ranges have been shown to exhibit smaller range shifts (*MacLean and Beissinger, 2017*; *Pinkert et al., 2022*), while increasing local temperature and loss of natural land cover may drive range retractions (*Pacifici et al., 2020*). However, exposure to extreme climate events (e.g. drought, heat waves, or storms) within species ranges may disrupt species' dispersal abilities and capacities to tolerate new conditions (*Kerr, 2020*; *Román-Palacios and Wiens, 2020*). Exposure to thermal anomalies can rapidly change entire communities and create shifts toward new ecosystems, sometimes leading to local declines (*Day et al., 2018*; *Grant et al., 2017*; *Harris et al., 2018*; *Román-Palacios and Wiens, 2020*).

While global change research on insects often emphasizes butterfly and bee taxa, recently assembled databases of odonate observations provide a rare opportunity to investigate species' spatiotemporal responses at larger taxonomic and spatial scales, particularly as most work has been done at national scales (*Córdoba-Aguilar et al., 2023*; *Kalkman et al., 2018*; *Sandall et al., 2022*). Due to their use of aquatic and terrestrial habitat across life different stages, dragonflies and damselflies are also considered indicator species for both terrestrial and aquatic insect responses to changing climates (*Hassall, 2015*; *Pinkert et al., 2022*; *Šigutová et al., 2025*), giving the study of these species broad relevance for conservation. There is some evidence that functional traits relate to odonates' interspecific variation in range shifts (*Angert et al., 2011*; *Grewe et al., 2013*), phenology shifts (*Diamond et al., 2011*; *Gutiérrez and Wilson, 2021*; *Zografou et al., 2021*), extinction risks (*Cardillo et al., 2008*; *Cooper et al., 2008*; *Rocha-Ortega et al., 2022*; *Suhonen et al., 2022*), and rates of decline and expansion within limited geographic scopes (*Powney et al., 2015*; *Rapacciuolo et al., 2017*; *Rocha-Ortega et al., 2020*). While relationships between morphological traits and range boundaries have been shown for some groups (i.e. *Rundle et al., 2007*), these may depend on species' geographic context. For example, differences in habitat connectivity and dispersal ability may constrain range shifts for lentic species (those species that breed in slow-moving water like lakes or ponds) and lotic species (those living in fast-moving water) in different ways (*Kalkman et al., 2018*). More southerly lentic species may expand their range boundaries more than lotic species, as species accustomed to ephemeral lentic habitats are better dispersers (*Grewe et al., 2013*), yet lotic species have also been found to expand their ranges more often than lentic species, potentially due to the loss of lentic habitat in some areas (*Bowler et al., 2021*). While warm-adapted species with more equatorial distributions could expand their ranges poleward following warming (*Devictor et al., 2008*), they could also increase in abundance in this new range area relative to species that historically occupied those areas and are less heat-tolerant (*Powney et al., 2015*).

In this study, we tested whether species with stronger geographical range shifts also advanced their emergence phenologies, or if one response offsets the need for the other. We also asked whether functional traits, range geography (i.e. southerly vs. northerly), or temperature variability predict range shifts at species' northern range limits, and whether these factors can also predict shifts in species emergence phenology. We predicted that species would exhibit shifts in both geography and phenology, as we expect species shifting in phenology to have larger populations, increasing likelihood of dispersal and successful range shifts. We predicted that species able to use both lentic and lotic habitats would shift their phenologies and geographies more than those able to use just one habitat type, as generalists outperform specialists as climate and land uses change (*Ball-Damerow et al., 2015*; *Ball-Damerow et al., 2014*; *Hassall and Thompson, 2008*; *Powney et al., 2015*; *Rapacciuolo et al., 2017*). Alternatively, species might respond to rapid climate changes in ways that reflect their geographical position, indicating that where they are found is a better predictor of their conservation risk from global change than their intrinsic biological characteristics.

## Results

### Relationship between range and phenology shifts

Most species (52 of 76) expanded their northern range limits toward higher latitudes (mean range expansions of 180 km). The average range expansion across all species was 63 km northward, although some species showed range retractions (*Figure 2*). Most species (41 of 66) maintained or advanced their emergence phenology (mean of –2.71 days in emergence phenology shifts among all species; *Figure 2*). Fewer species were included in phenology analyses than analyses of range shifts due to the

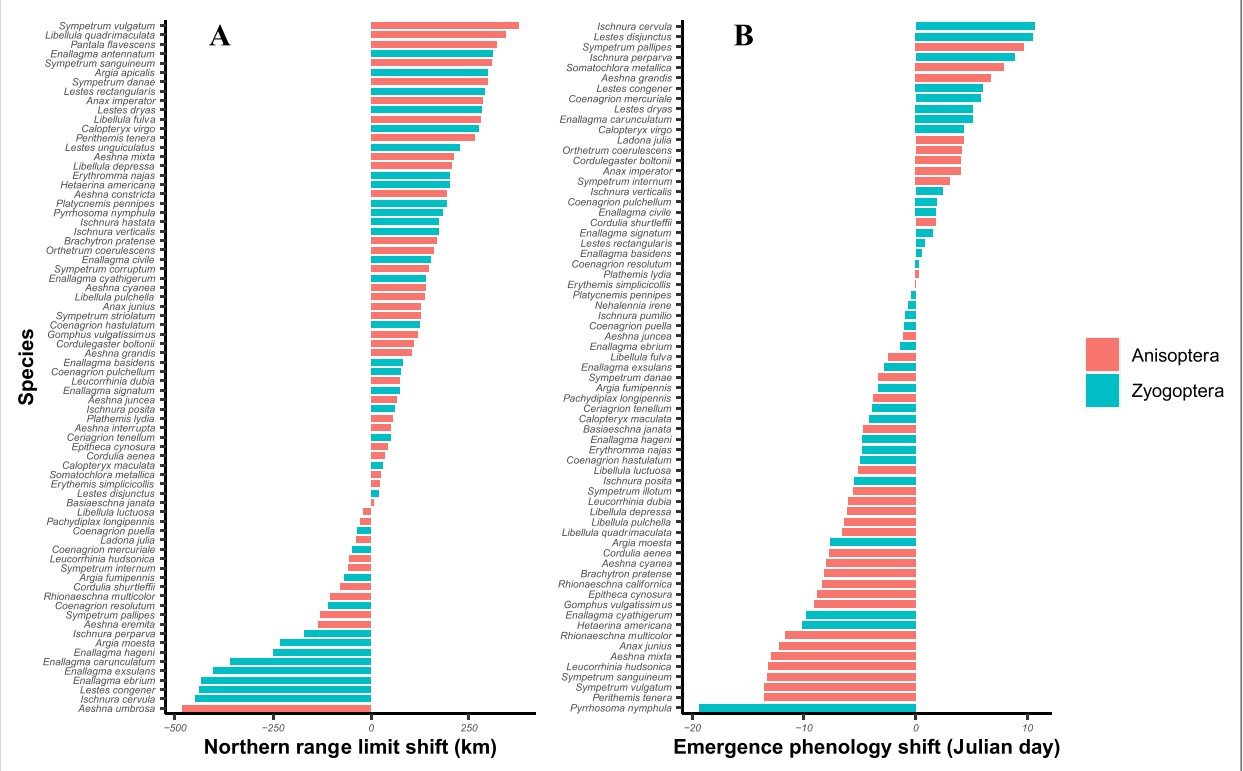

**Figure 2.** Distribution of northern range limit shifts (**A**) kilometers and emergence phenology shift (**B**) Julian day of 76 European and North American odonate species between a recent time period (2008–2018) and a historical time period (1980–2002). Anisoptera (dragonflies) are shown in pink, Zygoptera (damselflies) are shown in blue.

data intensity required to capture phenology shifts at a study site ($N = 66$ vs. $N = 76$). Many species (50%) showed both advancing emergence phenology and range expansions, while 10% of species showed neither range nor phenological shifts relative to historical baselines.

The effect of species' range shifts on phenology range shifts was significant in our model investigating the relationship between these responses, indicating that species shifting their northern range limits to higher latitudes also showed stronger advances in their emergence phenology (**Figure 3**). This result was consistent in generalized linear model (GLM) and Bayesian analyses (p < 0.01; **Table 1**) and was maintained across North America and Europe, with no effect of continent in the model. This trend was consistent among both dragonflies and damselflies, although there was considerable interspecific variation in the magnitude of spatial and temporal shifts. Accounting for phylogeny did not improve model predictions and did not explain greater model variance ($R^2$ = 17% and 15%, for GLM and Markov Chain Monte Carlo generalized linear mixed model [MCMCglmm] models, respectively).

## Drivers of range and phenology shifts

Range geography and climate variability, rather than functional traits, predicted range shifts in both North America and Europe, with range geography being consistently the strongest predictor. Species' functional traits did not relate to the extent of observed geographical range shifts in tests using GLM and MCMCglmm models. Species with more southern distributions shifted their northern range limits toward higher latitudes more than northern species or species present in both the north and south (**Table 2**; p = 0.002 and 0.004, model $R^2$=26.6% and 23.7%, for GLM and MCMCglmm models, respectively), with no effect of range size on range shifts. Species experiencing smaller changes in interannual temperature variability also had a higher likelihood of northern range limit shifts (p = 0.0005 for GLM, p = 0.002 for MCMCglmm). Results from the GLM and MCMCglmm models were qualitatively similar; however, a smaller amount of model variance was explained when phylogeny was accounted for. Emergence phenology shifts were not affected by species' traits, range geography, nor climate variability; due to this, model results are not displayed here. While range and phenology

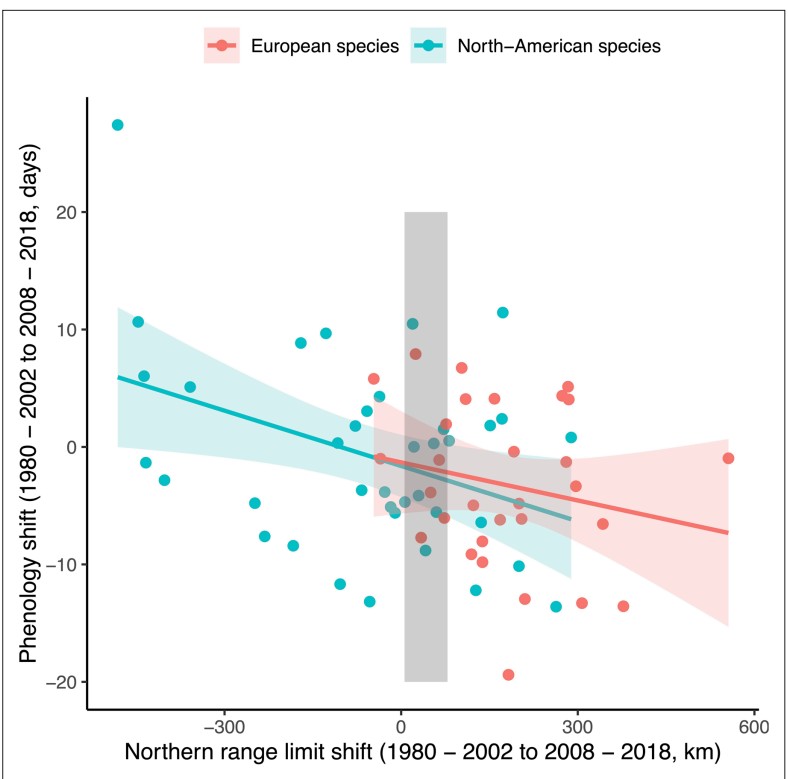

**Figure 3.** Relationship between range shifts and emergence phenology shifts among North American and European odonate species ($N = 66$; model $R^2 = 17.08$ for generalized linear model [GLM], 14.9% for Markov Chain Monte Carlo generalized linear mixed model [MCMCglmm]). For reference, the shaded area shows mean latitudinal range shifts of terrestrial taxa as reported by Lenoir et al. (calculated as the yearly mean dispersal rate of $1.11 \pm 0.96$ km per year over 38 years).

The online version of this article includes the following figure supplement(s) for figure 3:

**Figure supplement 1.** p-values and coefficients of 1000 generalized linear model (GLM) iterations testing whether range shifts as calculated from random datasets predict the range shifts measured in the study.

**Figure supplement 2.** Observed range shifts in km from the equator, against randomized predicted values according to four random datasets.

**Figure supplement 3.** p-values and coefficients of 1000 generalized linear model (GLM) iterations testing whether phenology shifts as calculated from random datasets predict the phenology shifts measured in the study.

**Figure supplement 4.** Observed phenology shifts in Julian day, against randomized predicted values according to four random datasets.

**Figure supplement 5.** Panels A and B show the trace and density estimates of a phylogenetic mixed effects model exploring the relationship between range and phenology shifts in North American and European odonates ($N = 66$).

**Figure supplement 6.** Panels A and B show the trace and density estimates of a phylogenetic mixed effects model testing whether ecological traits, and geographic and climatic attributes predict range shifts in North American and European odonates ($N = 76$).

response types are related, this suggests that the mechanisms underlying phenological shifts are different than those underlying range shifts.

A phylogenetic signal may indicate that there are traits that determine species' spatial and temporal responses to changing climate that were not measured in this study. Yet we detected no phylogenetic signal using Pagel's lambda or Blomberg's *K* in either geographical range or phenological responses (*Table 2*). Adding a phylogeny term to the MCMCglmm models also failed to produce a pattern different to the GLMs, and model performance did not improve when we accounted for phylogeny in our assessment of northern range shifts.

**Table 1.** Fixed effects estimates and associated statistics from the generalized linear model and generalized mixed effects model (accounting for phylogeny; for credible intervals, see Appendix 1—table 4) of the relationship between range shifts and emergence phenology change.

The continent term shows effects of the North American continent compared to the European continent as the reference level. *N* gives the number of species involved in the model, and an asterisk indicates statistical significance of the variable in question (p-value <0.05). The pseudo $R^2$ type is *Nagelkerke, 1991*.

**Phenology shift (*N* = 66)**

|  | GLM | | MCMCglmm | |
| --- | --- | --- | --- | --- |
| Predictors | Est. | p | Post.m | p |
| (Intercept) | 0.12 | 0.53 | 0.12 | 0.53 |
| Range shift | −0.45 | <0.01* | −0.45 | <0.01* |
| Continent | −0.22 | 0.44 | −0.22 | 0.44 |
| Model evaluation | | | | |
| AIC/DIC | 185.39 | | 185.43 | |
| Null model | 193.13 | | 193.12 | |
| Pseudo $R^2$ | 17.08% | | 14.90% | |

## Discussion

In one of the first studies to investigate both shifts of phenology and range at a continental scale, we find that dragonfly and damselfly species show pronounced geographical and phenological shifts that converged across Europe and North America. Species expanding their ranges poleward also emerged earlier in the spring on both continents (*Figure 3*), with shifts predicted by range geography and climate variability, but not functional traits. These results suggest that some species may have an advantage with respect to climate change: they demonstrate the flexibility to respond both

**Table 2.** Fixed effects estimates and associated statistics from the generalized linear model and generalized mixed effects model (accounting for phylogeny; for credible intervals, see Appendix 1—table 4) of drivers of odonate range shifts.

*N* indicates the number of modeled species, an asterisk indicates statistical significance of the variable in question, and a dash symbol shows that the variable was excluded from the final model. The pseudo $R^2$ type is *Nagelkerke, 1991*. For the categorical variables breeding habitat type and range geography, we used lotic habitat type and Northern range as reference levels, respectively.

**Range shift (*N* = 76)**

|  | GLM | | MCMCglmm | |
| --- | --- | --- | --- | --- |
| Predictors | Est. | p | Post.m | p |
| (Intercept) | −0.65 | 0.018 | −0.65 | 0.022 |
| Widespread distribution | 0.34 | 0.32 | 0.34 | 0.31 |
| Southern distribution | 0.95 | 0.002 | 0.95 | 0.004 |
| T° variability | −0.38 | 0.0005 | −0.38 | 0.002 |
| Model evaluation | | | | |
| AIC/DIC | 202.8 | | 202.9 | |
| Null model | 218.7 | | 218.7 | |
| Pseudo $R^2$ | 26.60% | | 23.70% | |
| Phylogenetic signal | | | | |
| Pagel's lambda (p) | 0.0057 (0.89) | | | |
| Blomberg's *K* (p) | 0.11 (0.47) | | | |

temporally and spatially to the onset of rapid climate change. Conversely, species that show neither geographic nor phenological shifts may be particularly vulnerable to climate change.

We found no evidence for a tradeoff between range and phenology shifts; instead, half of species shifted both range and phenology. Earlier seasonal timing allows species to stay within their climatic limits and maintain population growth rates (*Macgregor et al., 2019*), although earlier emergence could expose individuals to early season weather extremes (*McCauley et al., 2018*). As only a small proportion of odonate adults undertake long-range dispersal (*Conrad et al., 1999*), greater local population sizes should contribute to higher dispersal rates (*Mair et al., 2014*), facilitating range shifts (*Kerr, 2020*; *Leroux et al., 2013*). This is consistent with results from other taxa: among British butterflies, early emergence increased population growth and facilitated range shifts for species with multiple generations per year (*Macgregor et al., 2019*) Finnish butterfly species with the greatest population growth rates shifted both their phenology and ranges (*Hällfors et al., 2021*). Such population growth or maintenance, and therefore the potential for range shifts, is only possible if habitat is available (*Mair et al., 2014*). Future work should consider habitat availability alongside range and phenology shifts, as it may help explain why some species are able to shift their phenology but not their range.

Southern species were more likely to expand their ranges northward than northern species or species present in both the north and south. Species' ability to maintain large populations may be impaired in northern latitudes, where rates of climate change are high (*IPCC, 2021*), hindering dispersal and colonization that are precursors to range expansions (*Mair et al., 2014*). Further mechanistic understanding of these processes requires abundance data. Southern species may have narrower niche breadths than widespread or northern species and may respond more rapidly to climate change to track this narrower niche (*Hällfors et al., 2024*). Emerging mean conditions in areas adjacent to the ranges of southern species may offer opportunities for range expansions of these relative climate specialists, which can then tolerate climate warming in areas of range expansion better than more cool-adapted historical occupants (*Day et al., 2018*). Adaptive evolution and plasticity may enable high population growth rates in newly colonized areas (*Angert et al., 2020*; *Usui et al., 2023*), but this possibility can only be directly tested with long-term population trend data. While some species experienced range retractions, these may result from sampling variability or stochastic population fluctuations along the northern range edge.

Increasing frequency and severity of extreme weather limited species' geographical range responses (*Table 2*). This trend was independent of functional traits that are mechanistically linked to species' climate change responses, such as dispersal ability or habitat preference. Extreme temperatures can reduce population sizes, leading to local extinctions (*Román-Palacios and Wiens, 2020*), and reducing the likelihood of range expansions (*Mair et al., 2014*). In odonates, experimental evidence has demonstrated that larval mortality rises with short-term extreme weather (*McCauley et al., 2015*). Individuals that shift phenologies earlier in the season to avoid climate extremes could still be exposed to harmful conditions (*Iler et al., 2021*); for example, odonate populations that respond to unusually warm spring temperatures may experience high mortality if temperatures return to seasonal conditions. Species that experience extreme conditions may then be unable to successfully shift in time, reducing population sizes and reducing the likelihood of range shifts.

In contrast to previous work demonstrating that range and phenology shifts are at least partially determined by species traits (i.e. *Sunday et al., 2015*; *Zografou et al., 2021*), no functional trait, or combination of traits, explained these shifts in North American and European Odonata. While we could not capture all functional traits in this analysis, our results are consistent with other work that identifies climate velocity and sensitivity as the best predictors of range shifts and thermal preferences tracking in marine systems (*Pinsky et al., 2013*; *Schuetz et al., 2019*). Species' tolerances to increasingly variable temperatures also help to predict extinction risk during climate change (*Kerr, 2020*; *Rocha-Ortega et al., 2020*). The extent to which species' traits actually determine rates of range and phenological shifts, rather than occasionally correlated with them, is worth considering further, but functional traits do not systematically drive patterns in these shifts among Odonates in North America and Europe.

The geographic positions of species' ranges determine the local pressures and environmental factors to which they are exposed (*MacLean and Beissinger, 2017*; *Pacifici et al., 2020*), potentially masking or confounding the effects of traits that evolved under conditions determined by range

geography (*Schuetz et al., 2019*). This process could cause trait-related trends to differ across levels of biological organization (*Srivastava et al., 2021*), from local populations (where traits might be critical) to biogeographical extents (where traits might be unrelated to range or phenological shifts; *Grewe et al., 2013*; *Gutiérrez and Wilson, 2021*; *Sunday et al., 2015*; *Zografou et al., 2021*).

Given that species' functional traits did not predict temporal or geographic responses, it is unsurprising that species' responses were also independent of phylogenetic history (*Franke et al., 2022*). The phylogenetic approach did not improve model predictions in any model that we tested, and there was no phylogenetic signal in either response according to Pagel's lambda and Blomberg's $K$ (*Table 2*). These results are consistent with previous work that found no phylogenetic trend in local odonate population extinctions (*Suhonen et al., 2022*). There may be strong variation in thermal niches among closely related species: species that are geographically isolated adapt to different local climates, while species that co-occur may experience divergent selection within their climate tolerances (*Schuetz et al., 2019*).

It remains unclear if range and phenology shifts relate to trends in abundance, but our results suggest that there may be 'winners' and 'losers' under climate change (*Figure 2*). Climate 'winners', species that are shifting in space and time, may require more limited conservation intervention. Species expanding their ranges could be better supported if habitat area and connectivity are conserved, facilitating climate-driven range shifts (*Littlefield et al., 2019*). Species only shifting their phenologies may require further study, as phenology shifts may have positive or negative impacts on abundance (*Iler et al., 2021*). Climate 'losers', species that are failing to shift in both space and time, may require more direct conservation intervention, such as managed relocation (*Richardson et al., 2009*). Species that did not shift their ranges northwards or advance their phenology included *Coenagrion mercuriale*, a European species that is listed as near threatened by the IUCN Red List (*IUCN, 2021*), and is projected to lose 68% of its range by 2035 (*Jaeschke et al., 2013*). This group also includes *Coenagrion resolutum*, a common North American damselfly (*Swaegers et al., 2014*), for which we could not find evidence of decline. This may be due in part to the greater area of intact habitat available in North America compared to Europe, enabling *C. resolutum* to maintain larger populations that are less vulnerable to stochastic climate events. Still, this and other species failing to shift in range or phenology should be assessed for population health, as this species could be carrying an unobserved extinction debt. Our analysis of phenology and range shifts should be repeated in other taxa, as it may offer a method of identifying conservation actions among species groups.

Understanding how range and phenology shifts vary across species, and what drives this variation, is increasingly urgent as climate change alters local and regional environmental conditions. Here, we showed that odonate species exhibit convergent responses of range and phenology shifts across continents. While species with southern distributions were more likely to shift their ranges, increasing temperature variation limited geographical range responses among species in both Europe and North America. Climate change is associated with increasing variability as well as shifting mean conditions, contributing to species decline and even local extinction risks (*Duffy et al., 2022*). In this study, where species are found (i.e. their range geographies) determines whether they are exposed and respond to such negative pressures. Simultaneous consideration of shifts in range and phenology is a powerful and necessary approach to test aspects of species' vulnerabilities to rapid global changes. By considering both the seasonal and range dynamics of species, emergent and convergent climate change responses across continents become clear for this well-studied group of predatory insects.

## Materials and methods
### Biological records
We assembled ~2 million observations records for North American and European odonate species collected between 1980 and 2018. Data sources included online dataset aggregators GBIF (http://gbif.org/) and Canadensys (http://www.canadensys.net/), Odonata Central (*Abbott, 2006*), and other institutions (see Acknowledgments). While odonates were sampled opportunistically, biases associated with data that are not systematically collected are less likely to affect trends at large spatial and temporal scales, particularly if data are obtained from multiple independent sources (*Pyke and Ehrlich, 2010*; *Zattara and Aizen, 2021*). We removed records with incomplete or missing species identification, year, or locality information. We selected unique observations for species, location,

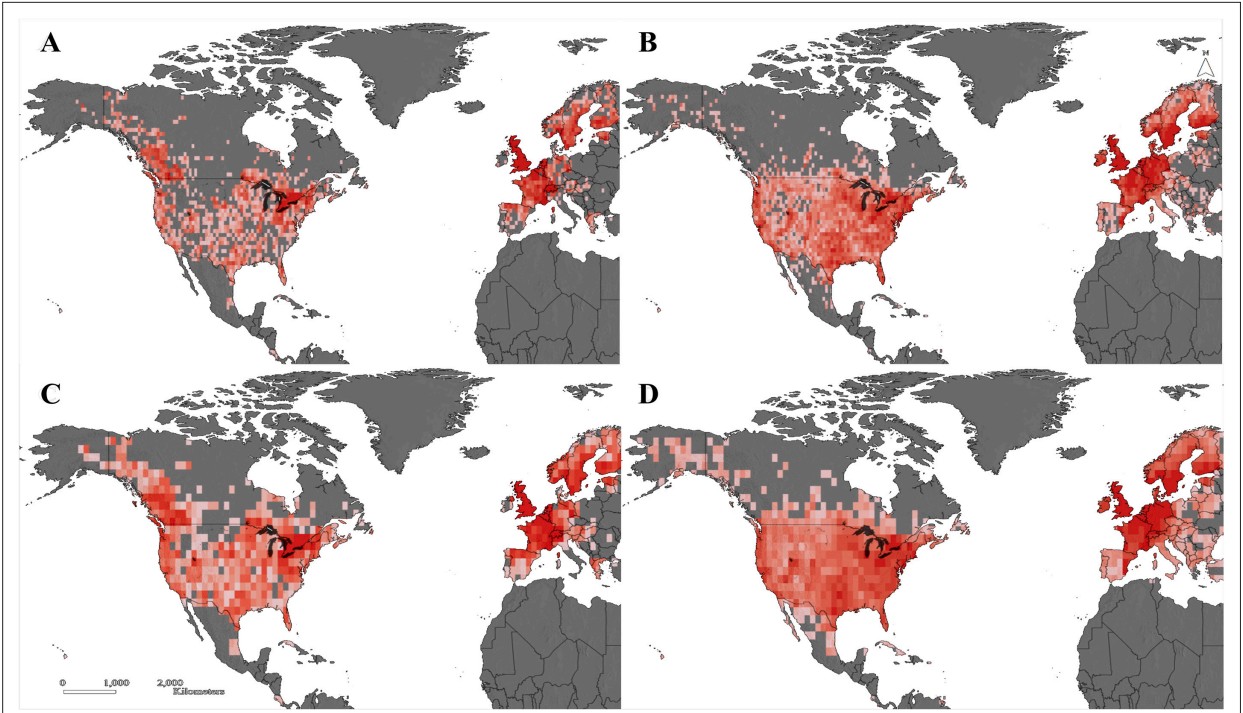

**Figure 4.** Richness of 76 odonate species sampled in North America and Europe in the historic period (1980–2002; panes A and C) and the recent period (2008–2018; panes B and D). Species richness per 100 × 100 km quadrat is shown in panes A and B, while panes C and D show species richness per 200 × 200 km quadrat. Dark red indicates high species richness, while light pink indicates low species richness.

year, and Julian day of collection and restricted the data to continental North America and Europe. We mapped species-specific observations using ArcGIS software (*ESRI, 2019*), and qualitatively verified species ranges. If a species was found on both continents, we only retained observations from the continent that was the most densely sampled. If we merged data for one species found on both continents, we could not perform a cross-continental comparison. However, if the same species on different continents was treated as different species, this would lead to uninterpretable outcomes (and the creation of pseudo-replication) in the context of phylogenetic analyses. In addition, species found on both continents did not have sufficient data to meet criteria for the phenology analysis.

We followed widely accepted methods to determine species range boundaries (*Devictor et al., 2012*; *Devictor et al., 2008*; *Kerr et al., 2015*), although other methods exist that are appropriate for different data types and research questions (*Ni and Vellend, 2021*). We assigned species presences to 100 × 100 km quadrats, a scale that is large enough to maintain adequate sampling intensity but still relevant to conservation and policy (*Soroye et al., 2020*), to identify the best sampled species. We excluded species found in fewer than 50 quadrats to increase the likelihood of accurately predicting the position of species' northern range boundaries. We retained ~1,100,000 records from Canada, the United States, and Northern Mexico, comprising 76 species (*Figure 4*). Observation records were separated into two time periods to compare species' recent phenologies and northern range limits (2008–2018) to conditions in a historical time period (1980–2002).

### Temperature variability

Temperature variability during a species' flight season may impact its ability to establish in new locations, or shift its emergence timing the following year. We downloaded a high-resolution gridded dataset for monthly average daily maximum temperature from the Climatic Research Unit (*New et al., 2002*). We extracted average values per 100 × 100 km quadrat across the months of April to October, covering the main flight period for odonates in this study, for each year included in the historical and recent periods. We calculated the coefficient of variation per quadrat for each time period and averaged these values per species and time period to measure interannual temperature variability

during species' flight season across their range. We used the difference between recent and historical measures as an estimate of change in temperature variability.

## Spatial and temporal change metrics

To limit potential effects of temporal and spatial biases, we generated range and phenology shift metrics with specific criteria for quadrats and species selection (*Bartomeus et al., 2019*; *Samy et al., 2013*; *García-Roselló et al., 2015*); we retained 76 species for range shift estimates (Appendix 1). Species' northern range boundaries were calculated using the mean of the 10 most northern points of each species range in both the historic (1980–2002) and recent (2008–2018) time periods, measured in kilometers from the equator (as in *Kerr et al., 2015*). We used the difference between range limit positions in the historic and recent time periods to estimate species' northern range limit shifts.

Since spatiotemporal biases sometimes inflate range shift measurements (*Kujala et al., 2013*), we used null models to test whether observed range shift estimates were robust, and the extent to which those responses differed from expectations arising because of rising sampling intensity over time. 1000 randomized datasets were created with the same number of species-specific geographical points per time period as the number of actual observations. Maximum and minimum latitude and longitude values were held constant relative to observed values. Northern range limit shifts were calculated as the difference in the mean of the 10 most northern points between the historic and recent time periods. We used GLMs (glm command in R) to test whether species-specific range limit shifts in each iteration predicted range shifts measured from the observation data (*Figure 3—figure supplement 1*). We found that observed northern range limit shifts are not consistent with expectations derived from changes in sampling intensity.

We estimated species-specific emergence phenology for each time period in 200 × 200 km quadrats; using larger quadrats increases probabilities of detecting signals of emergence phenology, which may otherwise be lost due to gaps in data density. We retained quadrats that contained at least 25 observations for a given species in both time periods. To estimate phenology per area, we used the *weib.limit* function of the *WeibullR* R package (*Pearse et al., 2017*). This function uses the Weibull distribution to estimate the Julian day of a species' first appearance and is especially useful to measure the timing of phenological events in sparsely sampled datasets. Techniques such as using the average of the $n^{th}$ first observations of phenological events, or the *n*th percentile flight dates (*Brooks et al., 2014*; *Robbirt et al., 2011*), tend to overestimate the timing of biological events due to temporal bias toward later days in the species' active period (*Pearse et al., 2017*). We retained emergence estimates between March $1^{st}$ and September $1^{st}$, as well as species and quadrats that showed a difference in emergence phenology of –25 to 25, –30 to 30, or –35 to 35 days between both time periods, to include only phenology shifts that could be biologically meaningful to environmental climate change. 68 species found across 63 quadrats met these criteria. Large changes in phenology are likely explained by other anthropogenic or natural factors or could occur due to noise in the data, since these phenology calculations per region are extremely data intensive. We calculated the difference in the day of emergence per quadrat between both time periods, as well as mean phenology change across all quadrats for each species. The number of quadrats per species used to calculate their mean phenological shift varied between 2 and 46.

We used null models to assess whether our approach to estimating phenology shifts was robust, as potential issues may arise due to spatiotemporal biases in the underlying data (*Kujala et al., 2013*). We constructed 1000 randomized datasets of species' hypothetical days of occurrence, using the same number of quadrats and observations within quadrats as in each time period of the observation records. We assigned the maximum and minimum Julian day of occurrence from the observation records to limit values in the randomized datasets. We applied the same method and criteria of inclusion to the randomized datasets as we did to measure phenology shifts from the observation data. GLMs were built to test whether phenology shifts calculated using 1000 random datasets predicted the phenology shifts that we measured. No discernible pattern emerged, indicating that observed shifts in phenology are not consistent with expectations derived from differences in sampling intensity over time (*Figure 3—figure supplement 3*).

## Range geographies and functional traits

To assemble trait data for the 76 species in the database, we used field guides (*Cannings, 2002*; *Jones et al., 2008*; *Paulson, 2012*) and existing trait databases (*Powney et al., 2014*; *Waller et al., 2019*). We considered any evolved morphological, physiological, behavioral, or life-history characteristic as a functional trait (*Beissinger and Riddell, 2021*). Geographic range and associated climatic characteristics are often considered ecological traits, as they are consequences of functional traits and their interactions with geographic features (*Bried and Rocha-Ortega, 2023*; *Chichorro et al., 2019*). Such ecological variables may predict species' responses to climate change and can add significant value to predictive models (*MacLean and Beissinger, 2017*). We identified whether species' ranges were more northern, southern, or both northern and southern (both), and determined the range size of each species by counting the number of quadrats occupied by that species in the historical time period.

Along with the geographic and climatic attributes, temperature variability and distribution, we selected four functional traits likely to be biologically relevant to spatial and temporal responses to climate change: flight duration, breeding habitat type (lotic, lentic, or both), egg-laying habitat (exophytic vs. endophytic), and body size (*Cannings, 2002*; *Jones et al., 2008*; *Paulson, 2012*; *Powney et al., 2014*; *Waller et al., 2019*). Species' flight period was measured as the total number of days of the flight period, estimated from the approximate time of the month of average first and last appearances. Breeding habitat was assigned according to a species' uses of lotic, lentic, or both habitat types. Egg-laying habitat was assigned according to whether species use exophytic egg-laying habitat (i.e. eggs laid in water or on land, relatively larger in number), or endophytic egg-laying habitat (i.e. eggs laid inside plants, usually fewer in number); species using exophytic habitats are associated with greater northward range limit shifts (*Angert et al., 2011*). Body size corresponded to the mean length of the abdomen of each species. We excluded overwintering stage and range size from our analysis as data were incomplete for many species, and excluded migratory behavior as the vast majority of species included in the study were non-migratory.

We tested for correlations among all predictors by calculating the Predictive Power Statistic (PPS) and Pearson correlations among traits (*Laken, 2021*): we found no evidence of correlation.

## Statistical analyses

We conducted statistical analyses using R Statistical Software (*R Development Core Team, 2019*). All continuous variables were transformed into Z-scores using the *scale* function in R. First, we investigated whether there was a relationship between species' range and phenological shifts by modeling phenology shift as the dependent variable, and range shift and continent as independent variables. We used both species-level frequentist (GLM; glm function in R) and Bayesian (MCMCglmm; *Hadfield, 2010*) models to improve the robustness of the results. We included a term to account for phylogeny in the MCMCglmm model, as species that are closely related are likely to have similar traits. We used the molecular phylogenetic tree published by the Odonate Phenotypic Database (*Waller et al., 2019*), which used a morphological and taxonomic phylogeny as the backbone tree, allowing species to move within their named genera or families according to molecular evidence (*Waller and Svensson, 2017*). Trace and density plots for the MCMCglmm model revealed no issues with autocorrelation or model convergence (*Figure 3—figure supplement 5*).

Next, we investigated whether functional traits, range geography, or temperature variability predicted range shifts at species' northern range limits, and whether the same predictors explaining range expansions could also predict shifts in species emergence phenology. We constructed two sets of GLMs, in addition to two sets of MCMCglmm accounting for phylogeny; one of each with changes in species' northern range limits as the response variable, and the other with changes in emergence phenology as the response variable. Non-significant variables, specifically all functional traits, were removed from the final geographic range shift model. No effects were significant in the model of phenology shifts. Trace and density plots for the MCMCglmm models did not indicate limitations related to autocorrelation or model convergence (*Figure 3—figure supplement 6*).

In addition to the inclusion of phylogeny in statistical models to account for potential data non-independence, we measured the phylogenetic signal in range and phenological shifts. We used the *phylosig* function of the *Phytools* package version 0.7-70 (*Revell, 2012*), which calculates phylogenetic signal using Pagel's lambda and Blomberg's *K*.

## Acknowledgements

The authors thank the anonymous reviewers and the handling editor for this paper, whose comments and suggestions helped improve the manuscript. We sincerely thank data contributors who have made this project possible: collections of the Royal Ontario Museum, the Royal BC Museum, and the California Academy of Sciences, as well as a repository of seven entomological collections of Québec (*Favret et al., 2020*). This research was supported by the Natural Sciences and Engineering Research Council of Canada (NSERC) through Discovery Grant and Discovery Accelerator Supplement funds to JTK, and research funds through the University Research Chair in Macroecology & Conservation from the University of Ottawa. CSD is grateful for the NSERC Alexander Graham Bell Canada Graduate Scholarship.

## Additional information

### Funding

| Funder | Grant reference number | Author |
| --- | --- | --- |
| Natural Science and Engineering Research Council of Canada | Discovery Grant and Discovery Accelerator Supplement | Jeremy Kerr |
| Natural Science and Engineering Research Council of Canada | Alexander Graham Bell Canada Graduate Scholarship | Catherine Sirois-Delisle |
| University of Ottawa | University Research Chair in Macroecology & Conservation | Jeremy Kerr |

The funders had no role in study design, data collection, and interpretation, or the decision to submit the work for publication.

### Author contributions

Catherine Sirois-Delisle, Conceptualization, Data curation, Formal analysis, Funding acquisition, Validation, Investigation, Visualization, Methodology, Writing - original draft, Project administration, Writing – review and editing; Susan CC Gordon, Formal analysis, Project administration, Writing – review and editing; Jeremy Kerr, Conceptualization, Resources, Software, Supervision, Funding acquisition, Validation, Writing – review and editing

### Author ORCIDs

Susan CC Gordon ⓘ https://orcid.org/0000-0003-0191-0359
Jeremy Kerr ⓘ https://orcid.org/0000-0002-3972-7560

Reviewer #1 (Public review): https://doi.org/10.7554/eLife.101208.4.sa1
Reviewer #2 (Public review): https://doi.org/10.7554/eLife.101208.4.sa2
Reviewer #3 (Public review): https://doi.org/10.7554/eLife.101208.4.sa3
Author response https://doi.org/10.7554/eLife.101208.4.sa4

## Additional files

### Supplementary files
MDAR checklist

### Data availability
Data are already published and publicly available, with those publications referenced in the methods section of the text.

The following previously published datasets were used:

| Author(s) | Year | Dataset title | Dataset URL | Database and Identifier |
|---|---|---|---|---|
| Waller JT, Willink B, Tschol M, Svensson EI | 2019 | Data from: The Odonate Phenotypic Database, a new open data resource for comparative studies of an old insect order | https://doi.org/10.5061/dryad.15pm5qc | Dryad Digital Repository, 10.5061/dryad.j1fd7 |
| Powney G, Brooks S, Barwell L, Bowles P, Fitt R, Pavitt A, Spriggs R, Isaac N | 2014 | British Odonata trait data | https://github.com/BiologicalRecordsCentre/Odonata_traits | GitHub, Odonata_traits |
| New M, Lister D, Hulme M, Makin I | 2002 | A high-resolution data set of surface climate over global land areas | https://crudata.uea.ac.uk/cru/data/hrg/ | Climate Research Unit, CRU TS v.4.05 |

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

# Appendix 1

## Odonate data

The full raw dataset includes ~2 million Odonata occurrence records. We first removed inadequate records from our primary dataset. Inadequate records were identified as those with incomplete information for species identification, year, or locality, inaccurate georeferenced points, and duplicate records. We retained ~1,100,000 unique species–location–date observations. There are 452,929 records in the first time period (1980–2002) and 641,590 observations in the second time period (2008–2018).

## Phenology and range position estimates

We have not tried to interpret noise in phenology shift estimates, which could occur due to sampling issues, or to another biologically meaningful eco-evolutionary response. Our methods do not enable tests of those ideas, however. For example, *Aeshna umbrosa* has a surprisingly high phenology shift (>25 days) but a very strong range retraction (<−300 km). For this species, we retained eight quadrats to calculate mean phenology shift estimate. It is possible that a section of the range was lost in which emergence was especially early, compared to other regions, due to local adaptations. This pattern may affect results where species appear to shift emergence dates earlier/later, but phenology shifts are affected by parts of the range that remain occupied.

We put in place criteria to make sure to include species with range shifts likely to result from climate change effects, rather than from sampling issues or to other anthropogenic pressures such as land use change or sudden land use intensification. *Nehalennia irene* was thus removed, as its range positions were highly unusual and likely due to other factors than global change (>800 km). *Libellula quadrimaculata* was removed due to sampling intensity discrepancies between time periods, having over 10,000 extra points in the second period compared to the first.

Temporal and spatial bias are likely to be present in opportunistic data, but they are less likely to impact long-term factors of species' distributions if including data from as many sources as possible, and that span across large geographic and temporal scopes (*Pyke and Ehrlich, 2010*; *Zattara and Aizen, 2021*). Here, we put in place several criteria in careful interpretation of results, as described in the Methods section of the main text. Further, among preliminary examinations of the data, we test for the effect of sampling onto our phenology and range shift estimates. We found no effect of sampling on range of phenology responses that we calculated here. We tested a linear mixed model with species as a random effect that assesses whether sampling intensity change per quadrat (where sampling intensity corresponds to sampling per species/quadrat) predicts species' phenology change per quadrat. The p-value of the predictor variable, sampling intensity change, was not significant. Secondly, we built a generalized linear model to assess whether greater sampling overall (measured as the number of quadrats in which each species is sampled) predicts species' range shifts that we computed. The p-value of the predictor variable, sampling change per species, was not significant. If spatial bias was present here, we may expect the number of quadrats in which species are found to increase the position of species' northern range boundaries.

## Model information and statements

We report the full model statements below as executed with the *MCMCglmm* R package, including the priors, and the model of trait evolution to account for phylogeny. Prior structure followed standard practice according to continuous or binary response variable. The family 'gaussian' or 'threshold' was selected depending on the continuous or binary nature of the data, respectively. The number of iterations, burnin, and thinning parameters were tested and confirmed with a visual assessment of model convergence using the trace and density plots. It is common practice for the burnin parameter to correspond to 10% of the number of iterations.

### Model 1

```
Aphylo ← vcv(phylogeny, model = "Brownian", corr = T)
    Ainv ← inverseA(phylogeny, nodes = "TIPS", scale = F)$Ainv
    prior ← list(R=list(V=1,nu = 0.002))
```

MCMCglmm(Phenological shifts ~Range shifts + Continent, ginverse = list(species = Ainv), family = "gaussian", prior = prior, DIC = T, nitt = 150,000, thin = 100, burnin = 15,000, data)

## Model 2
Aphylo ← vcv(phylogeny, model = "Brownian", corr = T)
 Ainv ← inverseA(phylogeny, nodes = "TIPS", scale = F)$Ainv
 prior ← list(R=list(V=1, nu = 0.002, fix = 1))
 MCMCglmm(Range shifts ~Breeding habitat type + Distribution region + Flight duration + Egg laying habitat + Body size+Temperature variability, ginverse = list(species = Ainv), family = "threshold", prior = prior, DIC = T, nitt = 150,000, thin = 100, burnin = 15,000, data)

## Model 3
Aphylo ← vcv(phylogeny, model = "Brownian", corr = T)
 Ainv ← inverseA(phylogeny, nodes = "TIPS", scale = F)$Ainv
 prior ← list(R=list(V=1,nu = 0.002))
 MCMCglmm(Phenology shift ~Breeding habitat type + Distribution region + Flight duration + Egg laying habitat + Body size+Temperature variability, ginverse = list(species = Ainv), family = "gaussian", prior = prior, DIC = T, nitt = 150,000, thin = 100, burnin = 15,000, data)

**Appendix 1—table 1.** Samples of 76 North American and European odonate species from between 1980 and 2018 followed our criteria for quality observation records for inclusion in our analysis of geographical shifts.

Species northern range limits (NRL) are shown in this table, as well as range limit shifts. All range limit values are shown in kilometers from the equator. We used the 10 most northern points of sampling in each time period to identify species' NRL, as detailed in the Methods section of the main text.

| Species | Continent | NRL (1980–2002) | NRL (2008–2018) | NRL shift |
|---|---|---|---|---|
| *Aeshna constricta* | North America | 5500.81 | 5692.6 | 191.79 |
| *Aeshna cyanea* | Europe | 6880.39 | 7018.26 | 137.87 |
| *Aeshna eremita* | North America | 7527.81 | 7394.66 | –133.16 |
| *Aeshna grandis* | Europe | 7504.84 | 7607.68 | 102.83 |
| *Aeshna interrupta* | North America | 7235.65 | 7286.83 | 51.18 |
| *Aeshna juncea* | Europe | 7747.56 | 7812.16 | 64.6 |
| *Aeshna mixta* | Europe | 6566.18 | 6776.3 | 210.12 |
| *Aeshna umbrosa* | North America | 6691.19 | 6209.63 | –481.55 |
| *Anax imperator* | Europe | 6263.92 | 6548.57 | 284.65 |
| *Anax junius* | North America | 5539.31 | 5666.11 | 126.8 |
| *Argia apicalis* | North America | 4922.17 | 5219.38 | 297.22 |
| *Argia fumipennis* | North America | 5235.2 | 5168.12 | –67.08 |
| *Argia moesta* | North America | 5533.74 | 5301.98 | –231.77 |
| *Basiaeschna janata* | North America | 5649.43 | 5655.77 | 6.34 |
| *Brachytron pratense* | Europe | 6737.42 | 6905.46 | 168.04 |
| *Calopteryx maculata* | North America | 5329.56 | 5359.13 | 29.57 |
| *Calopteryx virgo* | Europe | 7275.9 | 7549.35 | 273.45 |
| *Ceriagrion tenellum* | Europe | 5900.55 | 5950.63 | 50.08 |
| *Coenagrion hastulatum* | Europe | 7528.94 | 7651.51 | 122.57 |
| *Coenagrion mercuriale* | Europe | 5913.57 | 5867.06 | –46.52 |

*Appendix 1—table 1 Continued on next page*

*Appendix 1—table 1 Continued*

| Species | Continent | NRL (1980–2002) | NRL (2008–2018) | NRL shift |
|---------|-----------|-----------------|-----------------|-----------|
| *Coenagrion puella* | Europe | 6904.7 | 6869.58 | –35.12 |
| *Coenagrion pulchellum* | Europe | 7076.45 | 7152.81 | 76.36 |
| *Coenagrion resolutum* | North America | 7516.44 | 7408.7 | –107.73 |
| *Cordulegaster boltonii* | Europe | 7237.43 | 7346.93 | 109.49 |
| *Cordulia aenea* | Europe | 7400.12 | 7434.02 | 33.9 |
| *Cordulia shurtleffii* | North America | 7499.82 | 7421.98 | –77.84 |
| *Enallagma antennatum* | North America | 5037.73 | 5347.01 | 309.28 |
| *Enallagma basidens* | North America | 4768.61 | 4850.05 | 81.44 |
| *Enallagma carunculatum* | North America | 5950.86 | 5592.47 | –358.38 |
| *Enallagma civile* | North America | 5322.59 | 5473.78 | 151.19 |
| *Enallagma cyathigerum* | Europe | 7594.97 | 7733.06 | 138.09 |
| *Enallagma ebrium* | North America | 6341.31 | 5907.69 | –433.62 |
| *Enallagma exsulans* | North America | 5655.94 | 5253.94 | –402 |
| *Enallagma hageni* | North America | 6113.97 | 5865.52 | –248.45 |
| *Enallagma signatum* | North America | 5208.92 | 5281.09 | 72.17 |
| *Epitheca cynosura* | North America | 5541.19 | 5582.56 | 41.37 |
| *Erythemis simplicicollis* | North America | 5271 | 5292.65 | 21.65 |
| *Erythromma najas* | Europe | 7171.87 | 7372.13 | 200.25 |
| *Gomphus vulgatissimus* | Europe | 6890.76 | 7010.12 | 119.37 |
| *Hetaerina americana* | North America | 5044.41 | 5244.49 | 200.08 |
| *Ischnura cervula* | North America | 5973.57 | 5526.83 | –446.74 |
| *Ischnura hastata* | North America | 4725.07 | 4897.71 | 172.65 |
| *Ischnura perparva* | North America | 5608.92 | 5438.53 | –170.4 |
| *Ischnura posita* | North America | 5084.08 | 5143.79 | 59.71 |
| *Ischnura pumilio* | Europe | 6213.77 | 6769.62 | 555.85 |
| *Ischnura verticalis* | North America | 5579.68 | 5750.9 | 171.22 |
| *Ladona julia* | North America | 5908.68 | 5871.84 | –36.83 |
| *Lestes congener* | North America | 6240.48 | 5803.75 | –436.74 |
| *Lestes disjunctus* | North America | 7294.75 | 7314.26 | 19.51 |
| *Lestes dryas* | Europe | 6824.89 | 7108.33 | 283.44 |
| *Lestes rectangularis* | North America | 5263.85 | 5552.64 | 288.79 |
| *Lestes unguiculatus* | North America | 5803.91 | 6030.62 | 226.71 |
| *Leucorrhinia dubia* | Europe | 7675.72 | 7749.01 | 73.29 |
| *Leucorrhinia hudsonica* | North America | 7500.27 | 7446.85 | –53.42 |
| *Libellula depressa* | Europe | 6810.85 | 7015.37 | 204.52 |
| *Libellula fulva* | Europe | 6602.18 | 6882.5 | 280.32 |
| *Libellula luctuosa* | North America | 5337.86 | 5320.1 | –17.76 |
| *Libellula pulchella* | North America | 5556.31 | 5692.14 | 135.83 |

*Appendix 1—table 1 Continued on next page*

*Appendix 1—table 1 Continued*

| Species | Continent | NRL (1980–2002) | NRL (2008–2018) | NRL shift |
|---|---|---|---|---|
| *Orthetrum coerulescens* | Europe | 6744.55 | 6903.01 | 158.46 |
| *Pachydiplax longipennis* | North America | 5495.02 | 5467.11 | –27.92 |
| *Pantala flavescens* | North America | 5190.25 | 5510.15 | 319.9 |
| *Perithemis tenera* | North America | 4929.38 | 5192.47 | 263.09 |
| *Plathemis lydia* | North America | 5539.41 | 5594.8 | 55.38 |
| *Platycnemis pennipes* | Europe | 6934.22 | 7125.72 | 191.5 |
| *Pyrrhosoma nymphula* | Europe | 7131.07 | 7313.37 | 182.3 |
| *Rhionaeschna californica* | North America | 5641.53 | 5458.21 | –183.32 |
| *Rhionaeschna multicolor* | North America | 5581 | 5477.34 | –103.66 |
| *Somatochlora metallica* | Europe | 7718.67 | 7743.21 | 24.54 |
| *Somatochlora semicircularis* | North America | 6533.02 | 5994.13 | –538.89 |
| *Sympetrum corruptum* | North America | 5585.68 | 5733.2 | 147.52 |
| *Sympetrum danae* | Europe | 7268.54 | 7565.29 | 296.75 |
| *Sympetrum internum* | North America | 7179.45 | 7121.38 | –58.07 |
| *Sympetrum pallipes* | North America | 5655.65 | 5527.84 | –127.8 |
| *Sympetrum sanguineum* | Europe | 6663.43 | 6970.56 | 307.13 |
| *Sympetrum striolatum* | Europe | 6972.22 | 7098.74 | 126.52 |
| *Sympetrum vulgatum* | Europe | 6897.33 | 7274.78 | 377.45 |

**Appendix 1—table 2.** Sixty-six species sampled across North America and Europe between 1980 and 2018 followed our criteria for quality observation records for inclusion in our analysis of emergence phenology shifts.

Mean phenological shifts (PS) are measured in the number of Julian days comparing both time periods, as estimated using the Weibull distribution (see Methods). We also report the number of 200 × 200 quadrats used to calculate phenology estimates per species.

| Species | Number of quadrats | Mean PS |
|---|---|---|
| *Aeshna cyanea* | 37 | –8.05 |
| *Aeshna grandis* | 43 | 6.73 |
| *Aeshna juncea* | 34 | –1.11 |
| *Aeshna mixta* | 22 | –12.95 |
| *Aeshna umbrosa* | 8 | 27.42 |
| *Anax imperator* | 27 | 4.05 |
| *Anax junius* | 19 | –12.21 |
| *Argia fumipennis* | 8 | –3.39 |
| *Argia moesta* | 11 | –7.62 |
| *Basiaeschna janata* | 2 | –4.69 |
| *Brachytron pratense* | 24 | –8.18 |
| *Calopteryx maculata* | 15 | –4.14 |
| *Calopteryx virgo* | 41 | 4.35 |
| *Ceriagrion tenellum* | 14 | –3.88 |
| *Coenagrion hastulatum* | 20 | –4.97 |

*Appendix 1—table 2 Continued on next page*

*Appendix 1—table 2 Continued*

| Species | Number of quadrats | Mean PS |
|---|---|---|
| *Coenagrion mercuriale* | 11 | 5.80 |
| *Coenagrion puella* | 33 | −1.00 |
| *Coenagrion pulchellum* | 29 | 1.94 |
| *Coenagrion resolutum* | 2 | 0.33 |
| *Cordulegaster boltonii* | 34 | 4.08 |
| *Cordulia aenea* | 31 | −7.72 |
| *Cordulia shurtleffii* | 4 | 1.78 |
| *Enallagma basidens* | 2 | 0.52 |
| *Enallagma carunculatum* | 8 | 5.10 |
| *Enallagma civile* | 11 | 1.82 |
| *Enallagma cyathigerum* | 46 | −9.81 |
| *Enallagma ebrium* | 7 | −1.34 |
| *Enallagma exsulans* | 8 | −2.83 |
| *Enallagma hageni* | 5 | −4.79 |
| *Enallagma signatum* | 6 | 1.51 |
| *Epitheca cynosura* | 4 | −8.82 |
| *Erythemis simplicicollis* | 15 | 0.01 |
| *Erythromma najas* | 35 | −4.82 |
| *Gomphus vulgatissimus* | 12 | −9.14 |
| *Hetaerina americana* | 6 | −10.15 |
| *Ischnura cervula* | 3 | 10.65 |
| *Ischnura perparva* | 2 | 8.85 |
| *Ischnura posita* | 12 | −5.55 |
| *Ischnura pumilio* | 14 | −0.97 |
| *Ischnura verticalis* | 16 | 2.39 |
| *Ladona julia* | 9 | 4.29 |
| *Lestes congener* | 9 | 6.03 |
| *Lestes disjunctus* | 4 | 10.48 |
| *Lestes dryas* | 6 | 5.13 |
| *Lestes rectangularis* | 12 | 0.81 |
| *Leucorrhinia dubia* | 18 | −6.05 |
| *Leucorrhinia hudsonica* | 4 | −13.17 |
| *Libellula depressa* | 30 | −6.13 |
| *Libellula fulva* | 16 | −2.50 |
| *Libellula luctuosa* | 15 | −5.11 |
| *Libellula pulchella* | 14 | −6.43 |
| *Orthetrum coerulescens* | 22 | 4.11 |
| *Pachydiplax longipennis* | 13 | −3.84 |

*Appendix 1—table 2 Continued on next page*

*Appendix 1—table 2 Continued*

| Species | Number of quadrats | Mean PS |
|---|---|---|
| *Perithemis tenera* | 5 | −13.60 |
| *Plathemis lydia* | 14 | 0.30 |
| *Platycnemis pennipes* | 30 | −0.40 |
| *Pyrrhosoma nymphula* | 39 | −19.40 |
| *Rhionaeschna californica* | 2 | −8.41 |
| *Rhionaeschna multicolor* | 4 | −11.68 |
| *Somatochlora metallica* | 31 | 7.91 |
| *Sympetrum danae* | 32 | −3.34 |
| *Sympetrum internum* | 5 | 3.05 |
| *Sympetrum pallipes* | 3 | 9.67 |
| *Sympetrum sanguineum* | 28 | −13.30 |
| *Sympetrum vulgatum* | 16 | −13.56 |

**Appendix 1—table 3.** Ecological and geographical traits of 76 North American and European odonate species used in this work.

Field guides (*Cannings, 2002*; *Jones et al., 2008*; *Paulson, 2012*) and existing trait databases (*Powney et al., 2014*; *Waller et al., 2019*) were used to build this dataset. Habitat type represents species' breeding habitat and can have a value of lentic, lotic, or both types. Distribution shows the general geographic position of each species' range, which can be widespread (W), southern (S), northern (N), southern and widespread (S–W), or northern and widespread (N–W). Oviposition type corresponds to egg laying inside plants (endophytic) as opposed to directly in water or on plants (exophytic). Body size is measured as body length in mm. In the case that body length was given as a maximum and minimum value, we used the average of both values.

| Species | Habitat | Distribution | Flight | Oviposition | Body size |
|---|---|---|---|---|---|
| *Aeshna constricta* | Both | W | 2.5 | Endophytic | 69 |
| *Aeshna cyanea* | Lentic | S | 3 | Endophytic | 71.5 |
| *Aeshna eremita* | Both | N–W | 3 | Endophytic | 72.5 |
| *Aeshna grandis* | Both | N | 4 | Endophytic | 73.5 |
| *Aeshna interrupta* | Both | W | 3 | Endophytic | 66.5 |
| *Aeshna juncea* | Lentic | N | 4.5 | Endophytic | 75.5 |
| *Aeshna mixta* | Both | S | 2.5 | Endophytic | 60 |
| *Aeshna umbrosa* | Both | W | 3 | Endophytic | 68.5 |
| *Anax imperator* | Lentic | S | 2.5 | Endophytic | 76.5 |
| *Anax junius* | Both | S–W | 7 | Endophytic | 74 |
| *Argia apicalis* | Both | S–W | 4 | Endophytic | 36.5 |
| *Argia fumipennis* | Both | S–W | 3.5 | Endophytic | 31.5 |
| *Argia moesta* | Lotic | S–W | 3 | Endophytic | 39.5 |
| *Basiaeschna janata* | Both | W | 1 | Endophytic | 59 |
| *Brachytron pratense* | Lentic | S | 2 | Endophytic | 58.5 |
| *Calopteryx maculata* | Lotic | S–W | 3 | Endophytic | 48 |
| *Calopteryx virgo* | Lotic | W | 3.5 | Endophytic | 47 |

*Appendix 1—table 3 Continued on next page*

*Appendix 1—table 3 Continued*

| Species | Habitat | Distribution | Flight | Oviposition | Body size |
|---|---|---|---|---|---|
| *Ceriagrion tenellum* | Both | S | 3 | Endophytic | 30 |
| *Coenagrion hastulatum* | Lentic | N | 2.5 | Endophytic | 32 |
| *Coenagrion mercuriale* | Lotic | S | 2.5 | Endophytic | 29.5 |
| *Coenagrion puella* | Lentic | S | 3.5 | Endophytic | 34 |
| *Coenagrion pulchellum* | Both | S | 2 | Endophytic | 36 |
| *Coenagrion resolutum* | Lentic | N–W | 2.5 | Endophytic | 30 |
| *Cordulegaster boltonii* | Lotic | W | 3.5 | Endophytic | 77 |
| *Cordulia aenea* | Lentic | S | 2 | Exophytic | 51 |
| *Cordulia shurtleffii* | Both | N–W | 2 | Exophytic | 46 |
| *Enallagma antennatum* | Both | S–W | 2 | Endophytic | 30 |
| *Enallagma basidens* | Both | S–W | 4.5 | Endophytic | 24.5 |
| *Enallagma carunculatum* | Both | W | 3.5 | Endophytic | 31.5 |
| *Enallagma civile* | Both | S–W | 3.5 | Endophytic | 33.5 |
| *Enallagma cyathigerum* | Both | W | 3.5 | Endophytic | 32 |
| *Enallagma ebrium* | Both | N–W | 2.5 | Endophytic | 31 |
| *Enallagma exsulans* | Both | S–W | 3.5 | Endophytic | 34 |
| *Enallagma hageni* | Both | N–W | 2.5 | Endophytic | 30 |
| *Enallagma signatum* | Both | S–W | 2.5 | Endophytic | 32.5 |
| *Epitheca cynosura* | Both | S–W | 2.5 | Endophytic | 40.5 |
| *Erythemis simplicicollis* | Both | S–W | 2.5 | Endophytic | 41 |
| *Erythromma najas* | Both | S | 3 | Endophytic | 33 |
| *Gomphus vulgatissimus* | Lotic | S | 2 | Exophytic | 47.5 |
| *Hetaerina americana* | Lotic | S–W | 5 | Endophytic | 42 |
| *Ischnura cervula* | Both | W | 6 | Endophytic | 27.5 |
| *Ischnura hastata* | Lentic | S–W | 3.5 | Endophytic | 24 |
| *Ischnura perparva* | Both | S–W | 5 | Endophytic | 26.5 |
| *Ischnura posita* | Both | S–W | 3.5 | Endophytic | 25 |
| *Ischnura pumilio* | Lentic | S | 3 | Endophytic | 29 |
| *Ischnura verticalis* | Both | W | 3.5 | Endophytic | 26.5 |
| *Ladona julia* | Lentic | N–W | 2 | Endophytic | 41.5 |
| *Lestes congener* | Lentic | W | 2.5 | Endophytic | 36.5 |
| *Lestes disjunctus* | Both | W | 2.5 | Endophytic | 37.5 |
| *Lestes dryas* | Lentic | S | 2 | Endophytic | 37.5 |
| *Lestes rectangularis* | Both | S–W | 2.5 | Endophytic | 45 |
| *Lestes unguiculatus* | Both | W | 3 | Endophytic | 37.5 |
| *Leucorrhinia dubia* | Lentic | N | 2.5 | Exophytic | 33.5 |
| *Leucorrhinia hudsonica* | Lentic | N–W | 2 | Endophytic | 29.5 |
| *Libellula depressa* | Lentic | S | 2.5 | Exophytic | 43.5 |

*Appendix 1—table 3 Continued on next page*

*Appendix 1—table 3 Continued*

| Species | Habitat | Distribution | Flight | Oviposition | Body size |
|---|---|---|---|---|---|
| *Libellula fulva* | Lentic | S | 2 | Exophytic | 43.5 |
| *Libellula luctuosa* | Lentic | S–W | 2.5 | Exophytic | 46 |
| *Libellula pulchella* | Both | W | 2.5 | Endophytic | 54.5 |
| *Orthetrum coerulescens* | Lotic | S | 3 | Exophytic | 40.5 |
| *Pachydiplax longipennis* | Both | S–W | 3 | Endophytic | 35.5 |
| *Pantala flavescens* | Both | S | 3 | Exophytic | 48.5 |
| *Perithemis tenera* | Both | S–W | 4.5 | Exophytic | 22.5 |
| *Plathemis lydia* | Both | S–W | 3 | Exophytic | 45 |
| *Platycnemis pennipes* | Lotic | S | 2.5 | Endophytic | 36 |
| *Pyrrhosoma nymphula* | Lentic | W | 1.5 | Endophytic | 34.5 |
| *Rhionaeschna californica* | Lentic | S–W | 4 | Endophytic | 60.5 |
| *Rhionaeschna multicolor* | Both | S–W | 2 | Endophytic | 67 |
| *Somatochlora metallica* | Both | W | 1.5 | Exophytic | 53 |
| *Somatochlora semicircularis* | Lentic | W | 2 | Exophytic | 49.5 |
| *Sympetrum corruptum* | Both | S–W | 5 | Exophytic | 40.5 |
| *Sympetrum danae* | Lentic | W | 2 | Exophytic | 31.5 |
| *Sympetrum internum* | Lentic | W | 4 | Exophytic | 33.5 |
| *Sympetrum pallipes* | Both | S–W | 2 | Endophytic | 36 |
| *Sympetrum sanguineum* | Lentic | S | 3.5 | Exophytic | 36.5 |
| *Sympetrum striolatum* | Both | W | 5 | Exophytic | 39.5 |
| *Sympetrum vulgatum* | Lentic | S | 2 | Exophytic | 37.5 |

**Appendix 1—table 4.** Credible intervals of all MCMCglmm models testing predictions regarding the range and phenology shifts across 66 odonate species in North America and Europe.
These models are detailed in *Model information and statements* of the Supplementary Information.

| | Lower credible interval | Upper credible interval |
|---|---|---|
| **Phenological shifts ~ range shifts** | | |
| (Intercept) | –0.26 | 0.49 |
| Range shifts | –0.71 | –0.16 |
| Continent | –0.75 | 0.30 |
| **Range shifts ~ range geography + T° variability** | | |
| (Intercept) | –1.22 | –0.12 |
| Southern range | 0.36 | 1.56 |
| Widespread | –0.32 | 1.08 |
| T° variability | –0.60 | –0.18 |

