## [Editor Report · eLife Assessment]

The article presents **important** findings on the impact of climate change on odonates, integrating phenological and range shifts to broaden our understanding of biodiversity change. The study leverages extensive natural history data, offering a **convincing** analysis of temporal trends in phenology and range limit and their potential drivers.

---

## [Referee Report · Reviewer #1 (Public review)]

Summary:

This study evaluates whether species can shift geographically, temporally, or both ways in response to climate change. It also teases out the relative importance of geographic context, temperature variability, and functional traits in predicting the shifts. The study system is large occurrence datasets for dragonflies and damselflies split between two time periods and two continents. Results indicate that more species exhibited both shifts than one or the other (or neither), and that geographic context and temperature variability were more influential than traits. The results have implications for future analyses (e.g. incorporating habitat availability) and for choosing winner and loser species under climate change. The results also seem to support climate vulnerability assessments for species that rely on geographic range size and geospatial climate data layers rather than more detailed information (like demographic rates, abundances, or traits) that may not be so readily available. The methodology would be useful for other taxa and study regions with strong participatory ("citizen") science and extensive occurrence data.

Strengths:

This is an organized and well written paper that builds on a popular topic and moves it forward. It has the right idea and approach, and the results are useful answers to the predictions and for conservation planning (i.e. identifying climate winners and losers). There is technical proficiency and analytical rigor driven by an understanding of the data and its limitations.

---

## [Referee Report · Reviewer #2 (Public review)]

Summary:

This paper explores a highly interesting question regarding how species migration success relates to phenology shifts, and it finds a positive relationship. The findings are significant, and the strength of the evidence is solid. However, there are substantial issues with the writing, presentation, and analyses that need to be addressed. First, I disagree with the conclusion that species that don't migrate are "losers" - some species might not migrate simply because they have broad climatic niches and are less sensitive to climate change. Second, the results concerning species' southern range limits could provide valuable insights. These could be used to assess whether sampling bias has influenced the results. If species are truly migrating, we should observe northward shifts in their southern range limits. However, if this is an artifact of increased sampling over time, we would expect broader distributions both north and south. Finally, Figure 1 is missed panel B, which needs to be addressed.

Comments on revised version:

The revision has substantially improved the paper.

---

## [Referee Report · Reviewer #3 (Public review)]

Summary:

In their article "Range geography and temperature variability explain cross-continental convergence in range and phenology shifts in a model insect taxon" the authors rigorously investigate the spatial and temporal trends in the occurrence of odonate species and their potential drivers. Specifically, they examine whether species shift their geographic ranges poleward or alter their phenology to cope with changing conditions. Leveraging opportunistic observations of European and North American odonates, they find that species showing significant range shifts also exhibited shifts to earlier emergence. Considering a broad range of potential predictors, their results reveal that geographical factors, but not functional traits, are associated with these shifts.

Strengths:

The article addresses an important topic in ecology and conservation that is particularly timely in the face of reports of substantial insects declines in North America and Europe over the past decades. Through data integration the authors leverage the rich natural history record for odonates, broadening the taxonomic scope of analyses of temporal trends in phenology and distribution. The combination of phenological and range shifts in one framework presents an elegant way to reconcile previous findings and informs about the drivers of biodiversity loss.

Weaknesses:

To better understand whether species shifting both their ranges and phenology are more successful, or as stated here are 'clear winners', and hence whether those that do neither are more vulnerable would require integrating population trends alongside the discussed response. The ~10% species that have not shifted their distribution or phenology might have not declined in abundance, if they have rapidly adapted to local changes in climatic conditions (i.e. they might show a plastic response). These species might be the real 'winners', while species that have recently shifted their ranges or phenology may eventually reach hard limits. The authors are discussing this limitation but might want to adapt their wording, given the potential for misinterpretation. The finding that species with more northern ranges showed lesser northward shifts would speak to the fact that some species have already reached such a geographical range limit.

Achievements and impact:

The results support broad differences in the response of odonate species to climate change, and the prediction that range geography and temperature seasonality are more important predictors of these changes than functional traits. Simultaneously addressing range and phenological shifts highlights that most species exhibit coupled responses but also identifies a significant portion of species that do not respond in these ways that are of critical conservation concern. These results are important for improving forecasts of species' responses to climate change and identifying species of particularly conservation concern. Although not exhaustive regarding abundance trends, the study presents an important step towards a general framework for investigating the drivers of multifaceted species responses.

---

## [Author Response]

The following is the authors’ response to the previous reviews

**Reviewer #2 (Recommendations for the authors):**
Line 364-370: This paragraph is not very clear to me.

Thank you for pointing this out, we agree our point could have been made clearer. We have clarified as follows:

“The geographic positions of species’ ranges determine the local pressures and environmental factors to which they are exposed (MacLean and Beissinger, 2017; Pacifici et al., 2020), potentially masking or confounding the effects of traits that evolved under conditions determined by range geography (Schuetz et al., 2019). This process could cause trait-related trends to differ across levels of biological organization (Srivastava et al., 2021), from local populations (where traits might be critical) to biogeographical extents (where traits might be unrelated to range or phenological shifts; Grewe et al., 2013; Gutiérrez and Wilson, 2021; Sunday et al., 2015; Zografou et al., 2021).” (Lines 370-377).

**Reviewer #3 (Recommendations for the authors):**
L313: '...higher population growth' compared to what? Does this mean that species shifting to earlier emergence really show higher population growth over time?

Thank you for this suggestion, we have clarified as follows: “Earlier seasonal timing allows species to stay within their climatic limits and maintain population growth rates (Macgregor et al., 2019), although earlier emergence could expose individuals to early season weather extremes (McCauley et al., 2018).” (Lines 316-319).

L336: Same here. Please refer to your comparative counterpart in such statements. Does 'plasticity may enable higher population growth' mean higher than for species shifting range or phenology or higher compared to the previous level for a given species. In many cases it seems you are referring to an overall baseline, so that the 'higher' means 'lesser decline'. Wouldn't plasticity maintain population growth at similar levels as before? The current wording suggests that plasticity results in species exceeding their previous population growth. Please rephrase.

We agree it was confusing with no comparative counterpart, so we changed the sentence as follows: “Adaptive evolution and plasticity may enable high population growth rates in newly-colonized areas (Angert et al., 2020; Usui et al., 2023), but this possibility can only be directly tested with long term population trend data.” (Lines 341-343).

L307: The term 'universal winners' appears too strong and not well justified given the lack of the crucial third dimension of response. In fact, changes in phenology are less indicative than abundance trends. Combined with range shifts they would tell a story of success or failing, while phenological shifts would rather help to understand how species adapted. I am not saying the insight cannot stand alone, but it is important to adapt the wording in this regard.

Thank you for this comment, we have clarified the text as follows: “These results suggest that some species may have an advantage with respect to climate change: they demonstrate the flexibility to respond both temporally and spatially to the onset of rapid climate change.” (Lines 310-313).

We also softened language around winners and losers on line 388: “It remains unclear if range and phenology shifts relate to trends in abundance, but our results suggest that there may be ‘winners’ and ‘losers’ under climate change (Figure 4).” (Lines 387-388).

L326-240: I agree with line 330 that abundance trends are needed to clarify the situation of species shifting or not shifting ranges and phenology. However, this abstract should clarify that this is particularly important to understand whether non shifting species are really the 'losers'. If these species show adapted evolution or plasticity, we would expect they do not decline in abundance. Even without shifts in range or phenology they would be the 'ultimate winners' as you call it.

Thank you for this comment, we agree that abundance trends are necessary to understand potential winners and losers. We have made this addition to the abstract as follows: “Species shifting in both space and time may be more resilient to extreme conditions, although further work integrating abundance data is needed.” (Lines 16-18).